# Genomic and Phenotypic Characterization of CHO 4BGD Cells with Quad Knockout and Overexpression of Two Housekeeping Genes That Allow for Metabolic Selection and Extended Fed-Batch Culturing

**DOI:** 10.3390/cells14100692

**Published:** 2025-05-11

**Authors:** Nadezhda Alexandrovna Orlova, Maria Valerievna Sinegubova, Denis Eduardovich Kolesov, Yulia Alexandrovna Khodak, Victor Vyacheslavovich Tatarskiy, Ivan Ivanovich Vorobiev

**Affiliations:** 1Laboratory of Mammalian Cell Bioengineering, Institute of Bioengineering, Federal State Institution Federal Research Centre “Fundamentals of Biotechnology”, Russian Academy of Sciences, Leninsky Prospect, 33, build. 2, 119071 Moscow, Russia; nobiol@gmail.com (N.A.O.); mvsineg@gmail.com (M.V.S.); 52ru111@mail.ru (D.E.K.); salix33@gmail.com (Y.A.K.); 2Laboratory of Molecular Oncobiology, Institute of Gene Biology, 34/5 Vavilova Street, 119334 Moscow, Russia; tatarskii@gmail.com

**Keywords:** CHO, bak1, bax, apoptosis, autophagy, genome editing, CRISPR/Cas, bcl-2, beclin-1, dhfr, mammalian cell bioengineering

## Abstract

Re-engineering of CHO cells using genome editing and the overexpression of multiple helper genes is the central track for obtaining better cell lines for the production of biopharmaceuticals. Using two subsequent rounds of genome editing of the CHO S cells, we have developed the cell line CHO 4BGD with four knockouts of two pro-apoptotic genes bak1 and bax, and two common selection markers genes—glul (GS) and dhfr, and additional copies of genes bcl-2 and beclin-1 used for enhancement of macroautophagy. The NGS sequencing of 4BGD cells revealed that all eight targeted alleles were successfully disrupted. Two edited loci out of eight contained large inserts of non-relevant DNA. Further data analysis shows that cells have no off-target DNA editing events, and all known CHO genes are preserved. The cells obtained are completely resistant to the induction of apoptosis, and they are suitable for the generation of stably transfected cell lines with the dhfr selection marker. They also properly undergo the target gene amplification. The 4BGD-derived clonal cell line that secretes the monoclonal antibody retains the ability for prolonged fed-batch culturing. The method of obtaining multiply edited CHO cells using the multiplex CRISPR/Cas9 editing and simultaneous stable transfection of plasmids, coding for the housekeeping genes, is suitable for the rapid generation of massively edited CHO cells.

## 1. Introduction

Genetically re-engineered Chinese hamster ovary (CHO) cells are expected to allow better product titers due to longer culture durations in fed-batch mode, higher cell concentrations during the plateau phase, optimized protein biosynthesis pathways, restricted production of unwanted catabolites, and better fit to the metabolic selection of highly productive producer cells. Initial approaches to CHO cell engineering were based on chemical mutagenesis; the resulting *dhfr*^+/−^ and *dhfr*^−/−^ cell lines CHO DXB11 and CHO DG44 are widely used for biopharmaceutics production despite many irrelevant mutations in their genomes [1,2]. Targeted gene knockouts by zinc finger nucleases and transcription activator-like effector nucleases (TALEN)s [3] made it possible to obtain CHO derivatives with one inactivated gene [4] or two genes; the resulting cell lines, for example, CHOZN cells [5], were better suited for producer cell lines selection, but their use was restricted by patents and sophisticated licensing. The recent development of CRISPR/Cas genome editing provides for the relatively easy knockout of multiple genes in cultured cells [6], allowing researchers to generate many multiple knockout CHO genotypes in a few steps of editing and clone selections. The genome editing of CHO cells was additionally accelerated by the publicly accessible guide RNA (gRNA) database CRISPy, containing all possible gRNA candidates for the knockout of CHO genes by the Cas9 nuclease [7]. Clonal cell lines obtained using the CRISPR/Cas editing may be further optimized by overexpressing housekeeping genes and used for the development of new affordable biopharmaceutical expression platforms, tailored to specific culturing conditions and selection markers use.

The reduction of programmed cell death is an obvious and effective strategy for extending the culture duration in batch and fed-batch systems without compromising cell densities or specific productivity [8]. Sufficient blockade of mitochondria-induced apoptosis in cultured cells may be realized by inactivation or even the knockdown of only two genes coding the Bcl-2 homologs Bak1 and Bax [9]. Manipulations with other apoptosis-related genes either give less change in apoptosis suppression or are toxic for cells [10]. Bak1 and Bax double knockout in CHO cells leads to the beneficial cellular phenotype—better viability and extended productivity in fed-batch [11] and perfusion [12] processes.

We have hypothesized that the inactivation of Bak1 and Bax genes in CHO S cells can be performed together with the inactivation of *dhfr* and *glul* genes, allowing the efficient selection of stably transfected cells by plasmids with DHFR and GS selection markers in the quad-edited cells. The resulting cells may be further modified by overexpressing the macroautophagy inducer Beclin-1 and its natural inhibitor Bcl-2. The upregulation of autophagy in CHO cells by Beclin-1 and Bcl-2 overexpression was found to be beneficial for cell culture duration and integrated viable cell density [13].

We have attempted the CRISPR/Cas9 multiplex editing of four genes in CHO S cells*bak1*, *bax, dhfr*, and *glul*, and found one cell clone with seven inactivated alleles out of eight [14]. This clone, named 10/22 was subjected to the second round of genome editing and simultaneous transfection—with plasmids coding Bcl-2 and Beclin-1 under the control of *EEF1A1* (Eukaryotic Translation Elongation Factor 1 Alpha 1) promoter. The obtained clonal cell line with PCR-confirmed knockouts and good doubling time was named CHO 4BGD [15] and used for the creation of producer cell lines and phenotype characterization. The cell line developed was able to grow for 6 more days in batch culture conditions, making it the promising parental cell line for the production of biopharmaceuticals. Unlike the parental 10/22 cell line, it achieved the same peak cell density as intact CHO-S cells while maintaining longevity comparable to the 10/22 line (unlike CHO-S). Initial analysis of insertion–deletion mutations (indels) in the *dhfr* gene of the CHO 4BGD showed the presence of *E. coli*-derived DNA insert in the breakpoint, raising the question of the presence of other off-target genome modifications.

Genome editing techniques such as CRISPR/Cas9 have revolutionized the way we can modify CHO cells to enhance their productivity and functionality. However, genome editing can also result in unintended consequences. For example, off-target effects occur when the CRISPR/Cas9 system targets unintended sites in the genome, leading to unwanted mutations and potentially harmful effects [16].

Many studies have been devoted to the methods of reducing the number of off-target effects of Cas9, as well as to the methods of detecting such events in the genome [17]. Various cell culture-based methods have been developed to enrich the regions of interest; however, whole genome sequencing remains the gold standard. The problem of detecting off-target effects is especially acute in the case of genetically unstable immortalized cell lines, such as CHO cells [18].

Here, we present the results of the analysis of the whole genome sequencing data of edited CHO cells. Even relatively low coverage (10×) allowed exploration of target editing sites as well as off-target effects. An editing event was confirmed for each of the 5 g RNAs used, and foreign DNA insertion events in these regions were characterized. In addition, all sequences in the genome that differ from gRNA ≤ 4 bp were considered, and all possible events of non-targeted cutting of Cas9 were described. We have also confirmed the phenotype of CHO 4BGD cells by direct analysis of affected proteins and studied the cell behavior during the typical biopharmaceutical usage cycle—selection of stable transfectants, target gene amplification, and fed-batch culturing of clonal producer cell lines.

## 2. Materials and Methods

**Cell lines.** Obtaining cell line A11 with the inactivation of two *dhfr* alleles as well as line 10/22 with the inactivation of one *dhfr* allele and *bak1* and *bax* genes are described in the article [14]. Guide RNAs used for genome editing are present in Appendix A.

Obtaining of CHO 4BGD line with the homozygous knockout of *bak1*, *bax*, *dhfr*, *glul* genes and insertion of the expression cassettes of bcl-2, and beclin-1 genes are described in [15]. Briefly, the cells of 10/22 (*dhfr^+/−^ glul^−/−^ bak1^−/−^ bax^−/−^*) line [14] were transfected for *dhfr* inactivation with pKS-gD1plasmids encoding gRNA to *dhfr* gene, pX459ΔU6 encoding spCas9, and with two expression plasmids p1.2-Hygro-Beclin1 and p1.2-Zeo-Bcl-2 encoding for the additional copies of the *becn1* and *bcl2 C. griseus* genes under the control of *C. griseus EEF1A1* gene promoter, and the hygromycin and zeocin antibiotic resistance genes, respectively. Vector plasmids p1.2-Hygro (Addgene #162741, Watertown, MA, USA) and p1.2-Zeo (Addgene #162740) are described in [19].

Before transfection, the cells were cultured in ProCHO 5 medium (Lonza, Basel, Switzerland) with the addition of hypoxanthine-thymidine (Paneco, Moscow, Russia) and 4 mM alanyl-glutamine (Paneco) and transfected using Neon nucleofection device and kit (Thermo Fischer Scientific, Waltham, MA, USA) at 1700 V, 20 ms single impulse/After 48 h the medium was changed, puromycin was added in a final concentration of 14 µg/mL, and the cells were cultured for 3 days at 32 °C; the medium was then changed again, zeocin and hygromycin were added in final concentrations of 200 µg/mL and 500 µg/mL, respectively. Cloning was performed in ExCell CHO medium (Sigma, Livonia, MI, USA) supplemented with 8 mM glutamine and 2x HT.

**Polyclonal cell populations**, stably transfected with the plasmids coding for the SARS-CoV-2 S-protein RBD variants with the human IgG1 Fc tag (coded proteins RBDv2-Fc, RBDv3-Fc, plasmids p1.1-Tr2-V-RBDv2-sFc, p1.1-Tr2-V-RBDv3-sFc [20]), etanercept (fusion protein of the human tumor necrosis factor receptor 2 extracellular domain and the IgG1 Fc, TNFR2-Fc) and the chimeric antibody S25 toward botulinum neurotoxin A (IgG1—kappa) [21] were constructed as described in [20] with minor changes. Plasmids used for the transfection were assembled using the Chinese hamster elongation factor 1 alpha–based vector plasmids, derived from the p1.1-Tr2 plasmid (Addgene #162782) and contain the DHFR selection marker, linked to the genes of interest by the attenuated IRES. Plasmids, coding for the RBDv2-Fc and the RBDv3-Fc were transfected to the 10/22 cell line and two other plasmids, coding for TNFR2-Fc and S25 mAb—to the CHO 4BGD cells. Stably transfected cell populations were obtained using 50 nM MTX in the culture medium, and one round of target gene amplification was performed by increasing MTX to 500 nM and culturing cells for 20–30 days until their viability was restored to more than 85%. Target protein titers were measured for cell cultures with cell viabilities at more than 85% levels after the selection or amplification round. Cells were seeded as 300,000 cells per 1 mL of the ProCHO 5 medium, supplemented as described above, and cultured for exactly 3 days; the clarified culture supernatant was used for ELISA. Titers of target proteins were determined as the Fc-proteins, using the total human IgG kit (#271, Xema, Moscow, Russia) and the kit’s calibrator.

**Clonal producer cell line 4BGD-mAb** was obtained by nucleofection of the plasmid, based on the pCHO vector plasmid (Thermo Fischer Scientific, Waltham, MA, USA) and ORFs of the omalizumab antibody (humanized IgG1-kappa). The polyclonal cell line was obtained according to the vector plasmid manufacturer’s instruction, which was a two-step selection with MTX and puromycin. The resulting cell population was additionally genome-amplified by the 2 µM MTX treatment for 30 days. The resulting cell population was subjected to the cell cloning, described above. Three hundred forty microplate wells with single-cell colonies were used for the selection of best-producing clonal cell lines. The selected cell line with the highest specific productivity was re-adapted to suspension culturing and used for further experiments. The antibody concentration in the culture medium was determined by the ELISA kit (#271, Xema, Moscow, Russia).

**Control clonal cell line CHO S-hCG**, secreting the human chorionic gonadotropin, was obtained as described in [22], utilizing two steps of target gene amplification and the Chinese hamster elongation factor 1 alpha–based vector plasmids. The clonal cell line CHO S-hCG, used in cultivation experiments, was selected from at least 300 microplate wells with single-cell colonies. Hormone concentration in culture medium was determined using the corresponding ELISA kit (Xema, Moscow, Russia).

**A prolonged culture of CHO 4BGD** without changing the medium was performed in ProCHO 5 medium with 4 mM glutamine, 2xHT in 125 mL flasks, and a seeding density of 300,000 cells/mL. Samples were taken daily, and viable cell density was measured using a Countess II cell analyzer (Thermo Fischer Scientific, Waltham, MA, USA).

Prolonged culturing of the producer line without changing the medium was performed in the periodic fed-batch mode in 250 mL Erlenmeyer shake flasks (Corning, New York, NY, USA); culture volume was 60 mL, shaking frequency 155 rpm, 37 °C.

The 4BGD-mAb and CHO S-hCG cell lines were cultivated in Eden B600S Basal Medium using Eden F601aS Feed Medium and Eden F600bS Feed Medium (all from Bioengine, Shanghai, China), seeding density 500,000 cells/mL. Starting on day 3, 3% F601aS and 0.3% F600bS (by volume) were added every other day. Glucose levels were measured daily using an On Call Plus analyzer (Acon Laboratories Inc., San Diego, CA, USA); the glucose level in the culture was maintained in the range of 10–50 mmol with glucose added from 1 M stock solution if necessary. The pH number was adjusted to 7.0 using 1M NaOH when it dropped below the set value.

Cell lines 4BGD-mAb and CHO S-hCG were also cultivated in ActiPro Medium using CellBoost7a and CellBoost7b (all from HyClone, Logan, UT, USA). Starting on day 3, 1.75% CellBoost7a and 0.175% CellBoost7b F600bS (by volume) were added daily. Otherwise, the procedure was similar to the above-described culturing in Eden B600S medium.

**Genomic DNA analysis.** Genomic DNA was isolated with the ExtractDNA Blood & Cells kit of Evrogen and Wizard SV (Promega, Madison, WS, USA). DNA concentrations were measured with Qubit (Invitrogen, Carlsbad, CA, USA) and Qubit DNA Broad Range Assay Kit (Thermo Fischer Scientific, Waltham, MA, USA) for plasmid DNA or Qubit DNA High Sensitive Assay Kit for genomic DNA.

The whole-genomic library was prepared for the gDNA sample from the CHO 4BGD line; sequencing was performed with MGI DNPSEQ-400 (MGISEQ-2000) with an average coverage of at least 7× (2.1 × 10^10^ defined bases), PE 150 + 150 short paired-end reads; the files with paired-end reads in “fastq” format are available in the Sequence Read Archive (SRA) under the accession number SRR24907270 (BioProject: PRJNA983294).

The presence of integrated plasmids encoding Beclin-1 and Bcl-2 was confirmed using PCR with primer SQ-5CH6-F to the promoter region of the expression plasmids and the specific reverse primers AD-Beclin1-NheR or AD-Bcl2-NheR. For sequencing of the CRISPR edited sites, PCR with genomic DNA was performed using primers TA-DHFR-1v7-F and TA-DHFR-1v7-R, TA-GS-2v4-F and TA-GS-2v4-R, SQ-BAX456-F and SQ-BAX456-R, SQ-BAK467-1v7-F and SQ-BAK467-1v7-R (Table 1). PCR products were desalted with ethanol or isolated from the gel, sequenced using the Sanger method using the same primers and the sequence reads were used as reference sequences for assembly of NGS reads.

**RNA analysis.** RNA was isolated using a GeneJET RNA Purification Kit (Thermo Fisher Scientific, Waltham, MA, USA). The cDNA was obtained with a Mint-2 cDNA synthesis kit (Evrogen, Moscow, Russia) and quantified using a Qubit fluorometer (Invitrogen, Carlsbad, CA, USA) and the RNA BR kit (Invitrogen, Carlsbad, CA, USA). Quantitative PCR analysis was performed using an iCycler iQ thermocycler (Bio-Rad, Hercules, CA, USA) and qPCRmix-HS SYBR (Evrogen) reaction mixture with the primers shown in Table 1. 5–25 ng of cDNA were used for each PCR reaction. Amplification mode: pre-denaturation for 10 min at 95 °C, 40 cycles of amplification (10 s denaturation at 95 °C, annealing 15 s at 58 °C, elongation 15 s at 72 °C). Each sample was analyzed in at least three replicates. Two or three independent reverse transcriptions were performed. Threshold cycles and PCR efficiency, calibration curves, and copy number calculations were measured using the iCycler Iq4 software (Version 3.1., Bio-Rad, Hercules, CA, USA).

Two genes were used for normalization: *Mmadhc* (Metabolism Of Cobalamin Associated D), which is shown to be more suitable for comparison of the cells of different «ages», as it shows better expression stability [23], and *actb* (*actin beta*), a “traditional” housekeeping gene. The RT-PCR data were processed using the iCycler Iq4 software, including calculation of the reaction efficiency and threshold determination; qPCR raw data (Cq) were processed to get the data normalization and graphical representation using the web tool described in the paper [24] using the relative ΔΔCq method.

**Cytotoxicity analysis.** CHO-S and 4BGD cells were seeded in ProCHO 5 medium supplemented by 4 mM glutamine and the HT in 96-well plates (SPL Life Sciences, Pocheon-si, Republic of China) at 10 × 10^3^ cells/well, and drugs were added in two-fold dilutions starting from 50 μM. Cell death was induced with doxorubicin (Selleckchem, Houston, TX, USA), camptothecin, ABT-263 (ApexBio, Houston, TX, USA), and paclitaxel (Fujian South Pharmaceutical, Mingxi County, China). The cells were incubated for 72 h at 37 °C in 5% CO_2_, and then MTT (Sigma, Livonia, MI, USA) was added to a final concentration of 0.5 mg/mL for an additional 4 h of incubation at 37 °C in 5% CO_2_. After that, the media was carefully discarded, and cells were dissolved in 100 μL of DMSO (Paneco), and the optical density of the solution was measured at 570 nm wavelength using a Tecan Infinite spectrophotometer (Tecan, Männedorf, Switzerland). Effective concentrations of cytotoxic agents were used in cell cycle analysis.

**Cell cycle analysis.** CHO-S and 4BGD cells were seeded in ProCHO 5 medium supplemented by 4 mM glutamine and the HT in 10 cm Petri dishes (SPL Life Sciences, Pocheon-si, China) at 10 × 10⁶ cells/plate, and the drugs were added immediately after. Cell death was induced with 1 μM doxorubicin, 0.5 μM camptothecin, 1 μM ABT-263 and 0.5 μM paclitaxel. Cells were incubated for 48 h at 37 °C in 5% CO_2_, after which they were centrifuged at 300× *g* for 5 min, and the cell pellet was lysed in DNA-staining buffer (50 μg/mL propidium iodide (PI), 100 μg/mL RNAse A, 0.1% sodium citrate, 0.3% NP-40) for 30 min and analyzed using a Cytoflex S (Becton Dickinson, Franklin Lake, NJ, USA) flow cytometer in the Per-CP channel. Ten thousand events were collected for each probe. The data were analyzed using Cytexpert software (Version 2.6, Becton Dickinson, Franklin Lake, NJ, USA).

**Western blotting**: Whole-cell lysates were prepared with a modified RIPA buffer (50 mM Tris-HCl, pH 8,0; 1% Triton X-100; 0.1% Na-deoxycholate, 0.1% SDS, 150 mM NaCl, 1 mM EDTA, 0.01% NaN_3_) containing a protease inhibitor cocktail and 1 mM PMSF (both Sigma, Livonia, MI, USA). For all samples, total protein concentration was determined using the Bradford assay. The samples were normalized on total protein concentration, applied at 50 µg of total protein per lane, if not stated otherwise, and resolved on 12.5% SDS-PAGE gels followed by immobilization on a Hybond C Extra membrane (Cytiva, Marlborough, MA, USA) by the semi-dry transfer. Membranes were blocked by incubating with 1% non-fat dry milk (ECL blocking agent, Cytiva, USA) in Tris-buffered saline with 0.1% Tween-20 (TBST) for 1 h. The membranes were briefly rinsed two times by the TBST and probed with primary antibodies toward Bcl-2 (dilution 1:250; mouse monoclonal; Affinity Biosciences #BF9103, China), Beclin-1 (dilution 1:500; rabbit polyclonal; Affinity Biosciences #AF5128, Cincinnati, OH USA) and GAPDH (dilution 1:1000; mouse monoclonal; HyTest 6C5, Moscow, Russia); Bax (dilution 1:1000; rabbit polyclonal; Affinity Biosciences #AF0083, Cincinnati, OH, USA); Bak1 (dilution 1:500; rabbit polyclonal; Affinity Biosciences #DF6011, Cincinnati, OH, USA) in TBST with 1% non-fat dry milk for 1 h. The membrane was washed by TBST using two changes of washing buffer, one 15-minute wash, and two 5-minute washes. Incubation with anti-species antibodies-HRP conjugates was performed at room temperature for 1 h in TBST using anti-rabbit Ab (ABclonal AS014, Woburn, MA USA, dilution 1:1000) and anti-mouse Ab (Abcam ab6789, dilution 1:2000, Waltham, MA, USA). Membranes were washed as described above and developed with the luminol—H_2_O_2_ solution Pierce™ ECL Western Blotting Substrate (Thermo Fischer Scientific, Waltham, MA, USA) as per standard protocols. Original membrane images are present in Appendix A.

**Glutamine synthetase enzyme activity assay**. Five million live cells were collected by centrifugation, washed two times by the PBS at room temperature, resuspended in 250 μL of the ice-cold lysis buffer (100 mM Tris-HCl pH 8.0; 100 mM NaCl; 0.5% Triton X-100), lysed in Dounce homogenizer on ice, incubated for 5 more minutes on ice, centrifuged at maximum speed on a MiniSpin centrifuge (Eppendorf) for 2 min. The supernatant was centrifuged again for 5 min, and the residue was discarded. Total protein concentration was determined using the Bradford method. Twenty µL aliquotes of the samples (8–16 mg/mL total protein) and the blank sample, in triplicates, were mixed with 90 µL of the reaction solution, containing 270 mM Tris-HCl pH 7.2; 70 mM hydroxylamine-HCl pH 7.2, 35 mM MgSO_4_, 8 mM ATP, 170 mM sodium mono glutamate, and incubated at 37 °C for 3 h 40 min. Negative control reactions were run without the sodium mono glutamate substrate in the same conditions.

The reaction was stopped by adding to each test tube 37 µL of a stop solution consisting of equal volumes of 10% FeCl_3_ * 6H_2_O in 0.2 M HCl, 24% trichloroacetic acid, and 18.5% HCl solutions. Test tubes were mixed by vortexing and clarified by centrifugation for 5 min in a MiniSpin centrifuge (Eppendorf), max speed. Supernatants were incubated for 1 h at room temperature for complete color development. The optical density of the solutions was measured at 540 nm, and the specific activity of glutamine synthetase was determined as the optical density of the test tube minus the optical density of the negative control test tube, in mAU, divided by the total protein concentration, in mg/mL.

**NGS analysis.** The reads were aligned to the *C. griseus* genome (downloaded from Genbank as assembly GCF_000419365.1) using BWA (version 0.6.2) [25]. The resulting alignment files were processed using the set of utilities Samtools [26] and Bedtools [27], and visual evaluation was performed using the IGV genomic browser [28].

Samtools depth was used to calculate the coverage for the whole genome and plasmids. SNPs and indels were detected using bcftools mpileup [29]. RefSeq’s CriGri-PICRH-1.0 assembly annotation was used to identify the genes of interest and protein-coding sequences.

Off-targets were searched using Cas-OFFinder CRISPR RGEN Tools [27]. We were interested in all sequences in the genome different from cgRNA ≤ 4 bp, regardless of the position of the mismatch. The output was filtered, presented as a fasta file, and matched to the genome. To search for mutations/indels, we used bcftools calling, searched for intersections with exons from the genome annotation using bedtools intersect, and used bcftools mpileup to check for coverage in each region.

Several search options were required to find possible insertions of exogenous DNA into the genome of CHO cells, similar to the one that occurred in the case of Cas9 action to *dhfr*. We created a custom reference genome inclusive of CHO reference genome CriGri-PICRH-1.0 (GCF_003668045.3), 4 plasmids sequences (Appendix A), and *E. coli* strain TOP10 (GCA_019599065.1). Plasmids and *E. coli* genome sequences were appended to the CHO genome as fasta sequences, an extra chromosome. This custom CHOK1–Vector-*E. coli* reference genome is then used for aligning reads with bwa-mem aligner. Complete sequencing data have been deposited in the Sequence Read Archive (SRA) under the accession number SRR24907270 (BioProject: PRJNA983294).

The reads for which bwa-mem aligner found soft clipping were selected by bitwise filtering of the bam alignment file using samtools, or by using the extractSoftclipped toolkit (https://github.com/dpryan79/SE-MEI accessed on 5 May 2025) to create fastq files of the soft-clipped regions. In addition, the paired reads that related to different chromosomes were selected in a separate file, i.e., the pairs “CHO genome—plasmid” or “CHO genome—*E. coli*” were searched for.

The data on significant changes (complete removal of fragments, inserts with increased copying) were obtained by the following analysis of the original data. At the first stage, two fastq files were merged into one with the pairing information removed; then the resulting file was prepared: technical sequences, low-quality regions, and reads with the ultimate length below 50 nucleotides were removed (trimmomatic package). In the second stage, alignment was performed using the STAR package and the GriCri reference. Next, the coverage at each point was evaluated with the samtools depth package. At the final stage, a dedicated script was written: it identifies the sites with 0× or over 50× coverage and with lengths of 20 nucleotides or more (Appendix A). The resulting lists were analyzed using crossing bedtools intersections with the genome annotation list by gene. The resulting CDS lists were reviewed and analyzed manually.

**Statistical analysis** was performed using GraphPad Prism online calculators (https://www.graphpad.com/quickcalcs/ accessed 5 May 2025).

## 3. Results

The CHO 4BGD cell line was obtained by two sequential edits of CHO S cells using four gRNAs directed to the coding exons of the *dhfr*, *glul*, *bak1*, and *bax* genes near the start codons in the first round. After the first round of editing, a cell clone containing no intact alleles of the edited genes was detected by PCR and phenotypic testing [14]. This clonal line was found to contain one functional allele of the dhfr gene with a three-nucleotide insertion and a five-fold decrease in DHFR enzymatic activity.

The genome of this cell line was re-edited using another gRNA D1, also directed to the first coding exon of the *dhfr* gene. Simultaneously with the transfection of plasmids, which encoded gRNA’s and Cas9, the cells were transfected with plasmids encoding Bcl-2 and Beclin-1 of the Chinese hamster. Selection with hygromycin and zeocin antibiotics produced a population of stably transfected cells, which, by the method of limiting dilution, produced a clonal cell line CHO 4BGD with a doubling time of 28.5 h, containing no functional alleles of the *dhfr* gene according to the DHFR enzyme activity data and containing additional copies of Bcl-2 and Beclin-1 genes [15].

PCR technique was used to obtain amplicons of the DNA regions surrounding the PAM sites of the corresponding gRNAs. Their sequencing showed only a variant with 184 nucleotides insertion in the open reading frame (ORF) of the *glul* gene; for the *dhfr* gene, there were two different alleles, one with 5 nucleotides deletion and the other with three nucleotides insertion in the ORF and extensive insertion of *E. coli* DNA upstream of the start codon (Figure 1A). For the *bax* gene, this method detected only one variant with a single nucleotide insertion in the ORF; in the case of *bak1*, no PCR product was obtained, and the state of both alleles of the gene remained unclear. The presence of additional copies of the Bcl-2 and Beclin-1 genes and the corresponding selection markers in the cell genome was also confirmed by PCR.

### 3.1. NGS Analysis of CHO 4BGD Cells

A detailed analysis of the CHO 4BGD genome was performed to validate the inactivation of all alleles of the knockout genes, confirm the presence of correct extra copies of the Bcl-2 and Beclin-1 genes in its genome, and verify possible off-target gene modification events as well as non-specific gene damage and chromosomal rearrangements caused by nucleofection of the plasmid coding for the active nuclease.

We sequenced the genomic DNA of the CHO 4BGD line by the paired-end reads, with a target 10x read depth, and mapped the obtained data to the reference genome of *C. griseus* CriGri-PICRH-1.0 with the addition of p1.2-Zeo-Bcl2 and p1.2-Hygro-Beclin1 linearized plasmid sequences with overhangs of 75 nt. The BWA algorithm was used to align 99.69% of all reads, including secondary alignments, to this artificial genome. These secondary alignments were not excluded from subsequent analysis as they may correspond to genomic rearrangements or regions of the *C. griseus EEF1A1* gene from the Zeo-Bcl2 and p1.2-Hygro-Beclin1 plasmids.

To confirm heterologous DNA insertion events, we made separate fasta files consisting of 2 sequences: intact regions from the CHO genome and the same regions but with putative insertions. These files were used as references for additional alignment of the entire genomic dataset. The editing results were visually assessed in the IGV genome browser. Editing events were confirmed for all gRNAs, i.e., the reads with altered DNA sequences were detected for all edited loci, and no reads with unchanged DNA were observed. The *bak1* gene showed deletion of 8 nt and 10 nt in two alleles, and insertion of 1 nt for both alleles was observed in the *bax* gene (Figure 1A). The absence of both proteins in the 4BGD cells was confirmed by Western blotting (Figure 1B) of the 4BGD-derived clonal cell line.

Sequencing of PCR products has previously detected and now validated with NGS data the changes in both alleles of the *dhfr* gene at two points and confirmed the emergence of two different inactivated alleles containing multiple changes that we described earlier, including insertion events of *E. coli* DNA and DNA from plasmids used in genome editing. One of the *E. coli* DNA fragments contains a 4-nt region coinciding with 4 nt of the *dhfr* gene upstream of the break site; we expect that DNA recombination event was driven by this short direct repeat. Insertion of 184 nt of irrelevant DNA from chromosome 7 of the Chinese hamster and duplication of three nucleotides at the breaking point was also observed in one allele of the *glul* gene, which completely disrupted the open reading frame. Other disrupted allele of the *glul* gene was detected only by the NGS data analysis as the soft clipping event in one NGS read, where *glul* gene sequence in the gRNA binding area was connected to the DNA sequence, lying at the distance of 575 nt (edited2 allele of the glul gene on the Figure 1A). This finding was supported by the unusual distance between two reads in this paired read and, additionally, the presence of two more paired reads where first reads were mapped immediately upstream of the gRNA binding area in the *glul* gene and second reads were lying unusually distant from the first reads, approximate distance 1000 nt. The absence of functional GLUL (GS) enzyme in the CHO 4BGD cells was confirmed using the enzyme activity assay. Specific glutamine synthetase activity in cell lysates was measured using the ATP-dependent conversion of α-glutamate and hydroxylamine to γ-glutamylhydroxamate. This enzyme activity was at least 10 times lower in the CHO 4BGD cells than in the intact CHO S cells (2.99 ± 0.27 RU/mg total protein versus 30.40 ± 0.90 RU/mg total protein).

### 3.2. Analysis of Potential Off-Target DNA Editing Events

CHO 4BGD cell genome off-target editing events were considered for genomic loci in which gRNAs had homology of at least 16 nt out of 20, and the next 3 nucleotides corresponded to the acceptable PAM-site of Cas9 nuclease; a total of 468 loci for five gRNAs were observed in the CHO genome. For 411 loci, the average sequencing coverage was more than 5×; only 14 loci out of 411 had possible off-target changes, as shown in Table 2. Most of these possible events were single nucleotide substitutions; such mutations are not usually associated with Cas9 action. Nine single nucleotide mutations were detected at intergenic regions, and no mutations were detected in the open reading frames for 5 potentially mutation-affected genes. The data obtained allow us to determine that not a single gene was damaged by the off-target events.

We tried to find additional events of foreign DNA insertions in the CHO genome, analyzing the entire dataset for mapping anomalies. A total of 5,776,257 reads with paired-end reads corresponding to different chromosomes in the custom reference genome and 312,046 soft clipping events were observed. Positions in the CHO genome for which soft clips were detected were fixed using bedtools and visually inspected in the IGV genome browser tool. Over 90% of these unpaired reads and over 90% of the soft clips reads corresponded to regions of the *EEF1A1* gene included in the p1.2-Zeo-Bcl2 and p1.2-Hygro-Beclin1 plasmids used to overexpress the Bcl-2 and Beclin-1 genes. These pairs of reads and reads with soft clips were excluded from further analysis. Among the reads attributed to the *E. coli* genome, no reads were found to have a soft clip event with CHO chromosomes, except for the *E. coli* DNA insertions into the *dhfr* gene described above. Also, no soft clip events were detected for the reads aligned to the px458N and pKS-gRNA-BB plasmids, except for the px458N plasmid fragment insertion into the *dhfr* gene already described above. A total of 6286 and 561 reads were aligned on the px458N and pKS-gRNA-BB plasmid sequences. Thus, we cannot confirm any off-target or non-specific recombination events of px458N and pKS-gRNA-BB plasmid DNA or *E. coli* DNA in the genome of 4BGD cells.

Massive deletions in the genome of edited cells were searched as the genome regions of more than 20 nt long with zero coverage by the NGS reads for each nucleotide. Only 1805 such sites were found, and only two of them intersected with the coding sequences (CDS) of known genes—LOC113832938 and Sympk. Only two coding nucleotides of the Sympk gene have zero coverage by the NGS reads, so we expect that this absence of coverage was caused by the limited sequencing depth rather than the real deletion in the genome of the 4BGD cells. The cells developed have no detectable deletions in known genes.

### 3.3. Analysis of Genome-Integrated Plasmids and Expression of Plasmid-Encoded Proteins

The copy numbers of the p1.2-Zeo-Bcl2 and p1.2-Hygro-Beclin1 plasmids were determined by calculating the average read coverage of the open reading frame (ORF) regions of the target genes and the ORF regions of their corresponding selection markers. The results are presented in Table 3. The discrepancy in the calculated copy numbers for the ORFs of the target genes and selection markers can be attributed to the mapping of sequencing reads to either the host cell’s own genes or the plasmid sequences, which were added to the reference genome as additional chromosomes.

When the entire read pool was aligned to a reference consisting solely of the plasmids p1.2-Zeo-Bcl2 and p1.2-Hygro-Beclin1, the coverage of the open reading frames approximately doubled. This increase was due to the inclusion of reads corresponding to *bcl2* and *becn1* natural genes. Additionally, a significant number of clipped reads were observed at exon junctions. In contrast, when the read pool was aligned to a reference that included both the genome and the plasmids, no clipped reads were detected within the open reading frames of the target plasmid genes. This indicates that the BWA algorithm for paired-end reads effectively distinguishes between reads originating from genomic and plasmid genes. Consequently, the data obtained accurately represent the copy number of the target plasmids.

Functional activity of inserted plasmids was also assessed using Western blotting of CHO S, 10/22, and 4BGD cell lysates, all taken at the exponential growth phase at day 3 (Figure 1C). Knockout of Bak1/Bax in the 10/22 cell line resulted in significant downregulation of both Bcl-2 and Beclin-1. Subsequent transfection of additional Bcl-2 and Beclin-1 gene copies restored Beclin-1 level close to the initial value and only partially restored the Bcl-2 level (Figure 1D). Both proteins were significantly overexpressed in 4BGD cells versus their parental 10/22 cell line.

### 3.4. Phenotype of 4BGD Cells

We have tested the ability of the 4BGD cells to support the target gene amplification by culturing the cells, transfected by plasmids with the DHFR selection marker, in the presence of gradually increasing concentrations of the DHFR inhibitor methotrexate (MTX). The cells with the *dhfr*^−/−^ genotype, like the CHO DG44 cell line and, in some cases, cells with the *dhfr*^+/−^ genotype, like the CHO DXB-11 line, support this method of increasing the specific productivity, and the cells with the intact *dhfr* alleles are generally resistant to the MTX treatment. We used the parental cell line 10/22 with the *bak1*^−/−^*, bax*^−/−^*, glul*^−/−^ knockouts, and the *dhfr*^+/−^ one-allele knockout as the control, and have found out that only 4BGD cells allow the increase of the target protein titers during the one-step target gene amplification (Figure 1B) for a set of mutually similar Fc-fusion non-enzymatic proteins. Additionally, 4BGD cells gave much higher target protein titers after the initial selection of stably transfected cell population than the parental 10/22 cells.

The resistance of 4BGD cells to apoptosis induction by specific agents was studied using flow cytometry. It was found that 4BGD cells, in contrast to parental CHO S cells, are resistant to several apoptosis inducers—camptotothecin, doxorubicin, and ABT-263 (navitoclax)—but are sensitive to taxol (Figure 1C).

### 3.5. Fed-Batch Culturing of 4BGD -Derived Cell Line

Since the 4BGD cell line did not contain active alleles of *bak1*, *bax*, *dhfr*, and *glul* genes, and both plasmids encoding Bcl-2 and Beclin-1 were detected in its genome, this line was used to obtain the clonal producers of recombinant proteins. In particular, a clonal producer of omalizumab (humanized IgG1) with a specific productivity of about 10 pg/cell/day was obtained (4BGD-mAb); its growth properties and time course of mAb accumulation were compared with a clonal producer of human chorionic gonadotropin based on the intact CHO S cells. This clonal cell line has a similar specific productivity of about 10 pg/cell/day (CHO S-hCG) and a similar doubling time. It was found that when cultured in fed-batch mode in ActiPro medium (Figure 2A), 4BGD-derived cells accumulated the target protein for 17 days, and control CHO-S-based cells did so for 10 days; in both cases, the target protein stopped accumulating at the moment when cell viability dropped below 80% (Figure 2A,B). When cells were cultured in the more affordable Eden B600S medium in fed-batch mode, the culture duration decreased for both cell lines; however, the total culture time remained significantly longer for 4BGD derivatives (Figure 2C). Parental cell lines, CHO S and 4BGD, were also cultured in the Eden B600S medium with HT and glutamine, utilizing the same fed-batch protocol (Figure 2D). Parental CHO S cells demonstrated the same longevity as their descendant CHO S-hCG cells, and intact 4BGD cells retain viability above 80% for at least 19 days, 4 days longer than their descendant 4BGD-mAb cells.

### 3.6. Macroautophagy Markers Dynamics in 4BGD-mAb Cell Line

The expression dynamics of genes related to the induction of apoptosis and autophagy were studied using qPCR and western blot methods for cell samples obtained by culturing both producer cell lines in Eden B600S medium. According to the blotting data (Figure 2E), the level of Bcl-2 steadily increased during the fed-batch culture for both cell lines. Beclin-1 level decreased during the plateau phase in both cell lines. The level of the common autophagy marker, the lipidated form of the LC3 protein [30], seen on blot as the band with higher electrophoretic mobility (lower band, LC3-II), was dramatically increased in the 4BGD-mAb cells during the plateau phase, starting on day 9, also demonstrating the active process of macroautophagy in the 4BGD-mAb cells during the fed-batch culture. It should also be noted that the control CHO S cells, but not the 4BGD-mAb cells, demonstrated significant proteolysis of the Beclin-1 protein, seen right from day 4 as the minor 50 kDa band. This proteolytic breakdown of Beclin-1, caused by the caspase-3, is considered the mechanism of autophagy inhibition by the apoptosis process [31].

### 3.7. qPCR Analysis of Genes Involved in Apoptosis and Macroautophagy Development

We used the qPCR method to study the changes in the expression levels of several genes related to the processes of apoptosis and autophagy. The main internal apoptosis pathway in mammalian cells starts with the cytochrome c release from mitochondria to cytosol upon formation of the Bak1/Bax complex in the outer membrane of the mitochondria. The cytochrome c in the cytoplasm binds to the Apaf-1 protein [32], forming a di-heptameric complex—the so-called core apoptosome that recruits the procaspase 9, activates it to the active enzyme [33] and allows the activation by the caspase-9 of two effector caspases: −3 [34] and −7. Another protein, XIAP, counters the activation of the procaspase-9 both in the Apaf-1-dependent and -independent situations [35]. Additionally, the XIAP inhibits the activation of procaspase-3, preserves its association with the full apoptosome, and thus prevents both internal and receptor-induced apoptosis [36] (Figure 3A).

According to the RT-PCR data, the level of Apaf-1 mRNA in 4BGD-mAb cells is lower than in the control cells during the exponential growth phase; however, it rapidly increases in the plateau phase and during the culture decline (Figure 3B). The level of XIAP mRNA increases in the genome-intact CHO S-derived cells during culturing (Figure 3C). In the 4BGD-mAb cells, the XIAP mRNA level is only 42% of that of the CHO S cells at the exponential growth phase, and it remains low during the entire culturing.

Macroautophagy induction in cells can be traced by the changes in the expression level of many genes involved in the formation and growth of autophagosomes, summarized in Figure 4A. We have chosen four mRNAs, coding members of all three known protein complexes, which drive the key autophagosome formation events. The ULK1 kinase in the autophagy initiation complex is activated by the AMPK during the starvation and, in the absence of inhibition by the mTOR, phosphorylates the Beclin-1 (ATG-6) that leaves the complex with Bcl-2 inhibitor and forms the nucleation complex with Ambra1, ATG14 and the PI3 kinase VPS34. The third elongation complex is formed by the ubiquitin ligase-like enzymes ATG5, ATG7, ATG12, and ATG16. The E1-like enzyme ATG7 makes the first conjugate with LC3-I, which subsequently converts to the second LC3-I-ATG3 conjugate. At the same time, the ATG7 promotes the assembly of ATG16: ATG5-ATG12 enzyme complex, which converts the LC3-I-ATG3 conjugate to the autophagosome-binding lipidated LC3-II. ATG7 is expected to control the size of autophagosomes, and ATG14 — the number of autophagosomes [37] (Figure 4A).

We have screened four autophagy markers by RT-PCR and found that during the exponential growth phase, the ATG7 (Figure 4D) and ULK1 (Figure 4E), but not the ATG14 (Figure 4C) or Ambra1 (Figure 4B), are downregulated in the 4BGD-mAb cells. In the late culture of 4BGD-mAb cells, both Ambra1 and ATG14 are upregulated, the ATG7 remains at low levels, and ULK-1 is transiently upregulated at the maximal level on day 10. This expression pattern may be described as enhancement of autophagosome nucleation and absence of autophagosome elongation in the late culture of the 4BGD-mAb cells. Microscopy studies of the autophagosome numbers and size in the 4BGD-mAb and control CHO S-hCG cells may be the subject of further studies.

## 4. Discussion

Obtaining and using CHO cell lines with multiple genomic knockouts has become a common practice in recent years. The most obvious way to improve CHO cells for biopharmaceutical applications is to simultaneously increase their fed-batch culture time and remove metabolic selection marker genes from the cell genome. Using parental CHO S subline cells, apparently the best basic CHO cell line in terms of their ability to grow in high-density suspension culture [38], we performed multiplex editing of four genes by transfecting the cells with a set of four plasmids encoding 4 gRNAs and a plasmid encoding Cas9. Under these conditions, we easily detected a cell line containing changes in all alleles of the targeted genes and then, by re-editing the *dhfr* gene and simultaneously transfecting plasmids with the Bcl-2 and Beclin-1 genes, obtained a clonal cell line with complete inactivation of all alleles and overexpression of Bcl-2 and Beclin-1. By using the NGS analysis, we have confirmed the sequence of 4 various disrupted alleles, obtained previously by the Sanger sequencing of the PCR products, reconstructed the sequence of two more disrupted alleles, which were not detected by PCR, and corrected the sequence of double-edited allele of the *dhfr* gene, containing two adjacent insertions of the foreign DNA fragments.

We hypothesized that the simultaneous introduction of multiple gRNAs into cells could result in multiple off-target editing events as well as chromosome rearrangements; however, analysis of NGS data revealed only one plausible off-target editing event, a 6 nt insertion in the intergenic region and about 10 SNPs near potential off-target sites. None of these SNPs altered the open reading frames of known genes. Chromosome rearrangements or deletions of significant portions of the genome were also not detected. While this study evaluated off-target mutations, potential non-specific point mutations in 4BGD cells could not be comprehensively assessed due to the lack of precise genomic reference data for the parental CHO-S line. Nevertheless, CRISPR/Cas9 editing is widely regarded as highly specific, with non-specific mutations typically occurring at negligible frequencies. Conventional mutagenesis of CHO cells resulted in completely different levels of non-specific genome damage; for example, 259 indels in ORFs were found for the CHO DXB-11 cell line with inactivation of one *dhfr* allele, most of them with frameshifts [39].

Simultaneously, with the almost complete absence of mutations at off-target sites, we found several mutually independent events in the insertion of foreign DNA fragments into the targeted DNA sites. Fragments of the plasmid encoding Cas9 and a section of *E. coli* genomic DNA were found inside the disrupted genes. The genomic bacterial DNA most likely entered the cells as an impurity in the plasmid preparations. A similar phenomenon has been previously described for CHO cell lines stably transfected with plasmids, with the emergence of two *E. coli* proteins among CHO cell proteins being observed [18]. In our case, two short insertions of *E. coli* DNA were introduced, they do not correspond to the full ORF of *E. coli* proteins.

Complete disruption of the *dhfr* open reading frame in both alleles proved to be necessary for multi-stage amplification in the genome of transfected cells of expression genetic cassettes. Previously, the CHO DXB-11 cell line with the *dhfr*^+/−^ genotype was widely used for the industrial production of recombinant proteins. Amplification of target genes was usually performed with parallel culturing of many selected colonies of stably transfected cells, some of which increased productivity during prolonged treatment with increased concentrations of MTX. This method of amplification is very laborious and requires CHO culturing in adherent culture, usually with fetal calf serum; this method is highly unwanted for current good manufacturing practice.

Genome-edited CHO cells with knockouts of *bak1, bax*, and the initial *dhfr*^−/−^ genotype were previously shown to be ineffective for the MTX-driven selection process [40]. In our experiments, the 4BGD cells, but not the parent apoptosis-resistant *dhfr*^+/−^ cells 10/22, are suited for the MTX-driven selection and amplification; the resulting polyclonal cell populations have a specific productivity in the 5–15 pg/cell/day range, making them the proper candidate host for creation of industrial-grade clonal cell lines.

4BGD cells demonstrate selective resistance to apoptosis induction by known chemotherapeutic agents. Camptothecin, doxorubicin, and ABT-263 practically did not cause apoptosis in 4BGD cells, but taxol induced cell death in 4BGD cells almost as well as in parental CHO S cells. Such differences are caused by different mechanisms of their action—camptothecin is an inhibitor of topoisomerase I, doxorubicin—topoisomerase II, and ABT-263 is a direct inhibitor of anti-apoptotic proteins of the Bcl-2 family [41]. These agents directly activate Bak1 and Bax, or cause irreparable DNA lesions, resulting in the Bak1/Bax activation. Resistance of 4BGD cells to these apoptosis inducers confirms the absence of functional Bak1 and Bax in cells. Taxol is known to block microtubule depolymerization and cause the G2/M cell cycle arrest [42], which, in turn, may cause apoptosis by the Bak1/Bax-independent mechanism [43].

Two common apoptosis-related markers—Apaf-1 and XIAP, were significantly affected in 4BGD-derived cells. The Apaf-1 was upregulated during the late culture, whereas XIAP was downregulated. The absence of XIAP in the late 4BGD-mAb culture can describe the reason for massive 4BGD-mAb cell death after 19 days of the fed-batch culture in ActiPRO medium: the environmental stress can be a source of apoptosome-independent activation of the caspase-3 and was not countered by the downregulated XIAP. Further studies of the caspase activation during the natural or artificial environmental stress state are required for clarification of the rapid cell death phenomenon in very old cultures of the 4BGD-derived cells.

Recently, it was demonstrated that CHO DG-44 derived cell lines, secreting IgG and subjected to the fed-batch culturing in a bioreactor using a proprietary medium, developed no apoptosis markers during 14 days of culture. Their decline coincided with changes in parthanatos and ferroptosis marker levels—PAR, FSP1, and MDA accumulation and a decrease of GPX4 [44]. Apoptosis markers were present in these cell cultures starting at day 12, but only at elevated impeller speed, indicating that apoptotic cell death of stressed CHO cells is a major factor, limiting the length of the fed-batch culture and product titers achieved.

During fed-batch cultivation of the 4BGD-mAb, we observed that it responded to environmental stress with increased LC3 protein lipidation, which directly indicates the induction of macroautophagy in these cells. Both 4BGD-mAb and CHO S-hCG cells showed some Beclin-1 decline, then transited from exponential growth to plateau phase. At the same time, both cell lines gradually increased Bcl-2 expression levels during the fed-batch run. Apparently, such upregulation of Bcl-2 corresponds to the usual response of CHO cells to increasing environmental stress [45]. Recently, it was shown that the overexpression of Bcl-2 in CHO S cells is insufficient for the extension of the fed-batch culture to a significant extent [46]. This observation is in line with our data, demonstrating that simple constitutive overexpression of anti-apoptotic proteins is not enough for the prevention of apoptosis in late-fed-batch culture, utilizing modern CHO culture media and feeds.

It was already shown that macroautophagy could be induced in *bak1*^−/−^, *bax*^−/−^ cells in the same way as in intact cells, and the Bcl-2 level during the induction of autophagy by the ABT-737 agent was not changed in *bak1*^−/−^, *bax*^−/−^ cells [47]. In our study, we investigated the test cell line and control cell line, both secreting large amounts of proteins, during the fed-batch cultivation in shake flasks, which closely mimics the bioreactor run. In these conditions, the 4BGD-mAb cells demonstrated a rise in both Ambra1 and ATG14, indicating the upregulation of the VPS34 complex I formation and subsequent enhancement in autophagosome formation [48]. The level of ULK1 kinase, which triggers the Beclin-1 activation, was initially decreased in 4BGD-mAb cells but increased over time, peaking at day 10. A comparative study, involving various cell types but all with wild-type Bak1 and Bax, shows that rapid induction of autophagy by amino acid starvation leads to upregulation of ATG14 and ULK1 [49]. In our case, the ULK1 level increased in late culture but was low during the exponential growth phase, so our results are generally in line with previous observations. The expression pattern of Ulk1 and ATG7 was also previously studied for the CHO cells with overexpressed Bcl-2 and Beclin-1 [13]. During the exponential growth phase, cells showed a decrease in Ulk1 expression, similar to our observations, and in the late culture of both the Bcl-2 -overexpressing cell line and the Bcl-2 + Beclin-1 overexpressing cell lines, the Ulk-1 was upregulated. The ATG7 level was similar in the Bcl-2 + Beclin-1-overexpressing cells and the control cells; these results are inconsistent with our data and may be explained by the knockout of Bak1 and Bax in the 4BGD-mAb cells.

Artificial rapid induction of the macroautophagy by amino acids withdrawal and artificial ER stress induction by hyperosmolar solution or sodium butyrate can dissect the specific macroautophagy initiation and general response to environmental stress in the 4BGD cells, thus being the subject of further studies.

## 5. Conclusions

Multiplex editing of the CHO cell genome using CRISPR/Cas9 revealed several events of insertion of extended foreign DNA at the editing points. Fragments of *E. coli* genomic DNA, a fragment of CHO genomic DNA, and a fragment of the expression plasmid encoding Cas9 were observed among the inserted DNA. While the insertion of *E. coli* genomic DNA can be prevented by increasing the purity of the plasmids isolated for transfection, events of cell genome fragment transfer and insertion of plasmid fragments used for cell transfection cannot be prevented using this method of genome editing. Simultaneous knockout of many genes and overexpression of at least two genes can be performed in one or two transfection rounds using plasmid sets encoding the corresponding gRNA and plasmids encoding the nuclease Cas9 with the puromycin resistance gene and plasmids that encode overexpressed genes.

Changes in the expression levels of genes related to the induction of apoptosis and macroautophagy observed during culturing of the 4BGD-derived cells show that blocking the internal pathway of apoptosis alters the process of macroautophagy induction. 4BGD cells do not contain extrinsic mutations that disrupt the functioning of known genes. They may be used for the development of high-yield producer cell lines using DHFR selection marker and MTX-driven target gene amplification. The resulting clonal cell line, secreting humanized mAb, retains the ability to grow in a fed-batch culture for a prolonged period.

## Figures and Tables

**Figure 1 cells-14-00692-f001:**
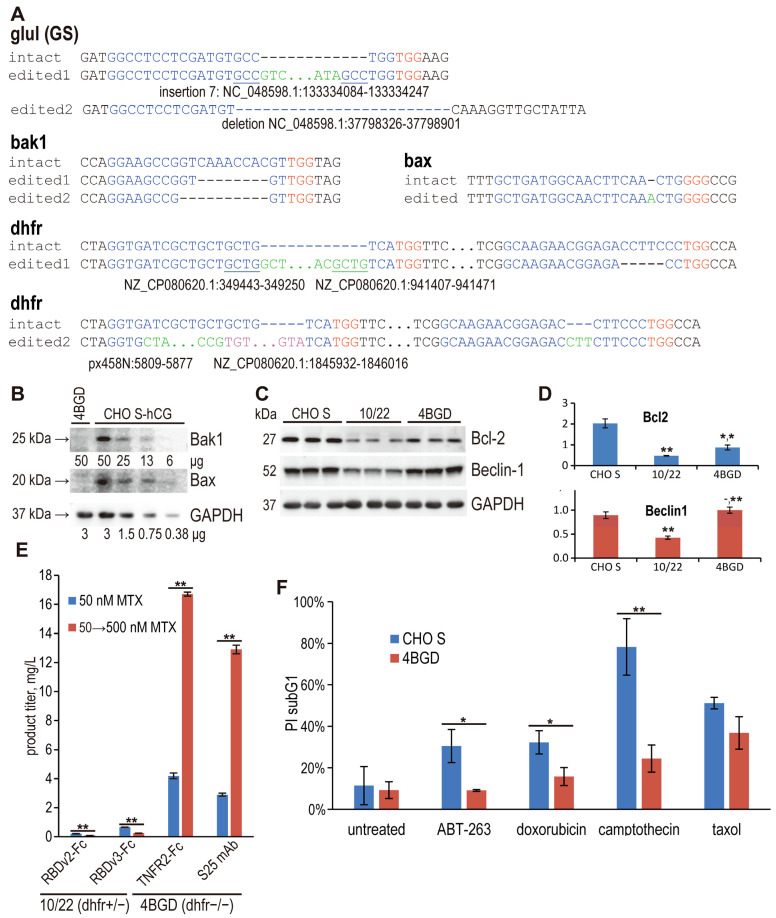
Alignments of the edited genomic loci in the CHO 4BGD cells, according to the NGS data, analysis of Bak1, Bax levels, changes in the Bcl-2 and Beclin-1 expression levels, changes in the cell’s productivities during the target genes amplification attempts and induction of apoptosis. (**A**)—alignments of the edited loci, guide RNA’s sequences are in blue, PAM sites—in red, inserts—in green and in violet. Direct repeats are marked by underlining. Coding strands are shown as plus strands for all loci. The sequence of the plasmid px458N is present in Appendix A, Genbank entry NC_048598.1 is the CHO chromosome 7; NZ_CP080620.1—*E. coli* TOP10 strain. (**B**)—Western blotting of cell lysates with anti-Bak1 and anti-Bax antibodies, day 3 samples from Figure 2D are used. (**C**)—Western blotting of cell lysates with anti-Bcl-2 and anti-Beclin-1 antibodies, untransfected cells, day 3 samples were used. Three replicate samples for each cell line were tested. (**D**)—change in Bcl-2 and Beclin-1 expression levels determined by Western blotting. (**E**)—change in target protein titers upon target gene amplification by increasing the MTX concentration in culture medium from 50 nM to 500 nM. Product titers are measured on day 3 of cultures. Cells were seeded as 3 × 10^5^ cells/mL. (**F**)—induction of apoptosis in the 4BGD cells by various agents, measured by flow cytometry, propidium iodide staining. (**D**)—Error bars indicate the standard deviation, *n* = 3 for panel (**D**), *n* = 2 for panel (**E**), *n* = 3–4 for panel (**F**). Statistical significance of differences by the two-tailed *t*-test, *—*p* < 0.05, **—*p* < 0.01.

**Figure 2 cells-14-00692-f002:**
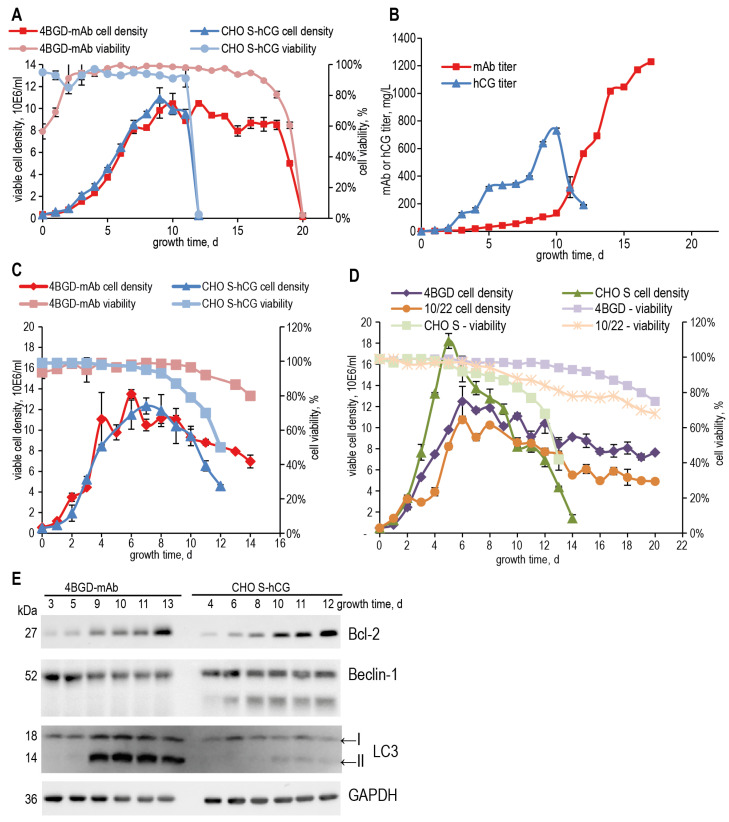
Fed-batch culturing of the 4BGD-mAb, CHO S-hCG, CHO 4BGD, and CHO S cell lines and western blot analysis of the Bcl-2, Beclin-1, and LC3 isoforms expression. Cell lines 4BGD-mAb and CHO S-hCG were cultured in ActiPRO (**A**), and Eden B600S (**C**) media, and target protein concentrations were determined for cultures in ActiPRO medium (**B**). (**D**)—parental cell lines CHO S, 10/22 and 4BGD were cultured in Eden B600S medium. Error bars indicate the standard deviation, *n* = 2. (**E**)—Western blotting for cell samples, taken during the cultivation in Eden B600S medium, depicted on (**C**).

**Figure 3 cells-14-00692-f003:**
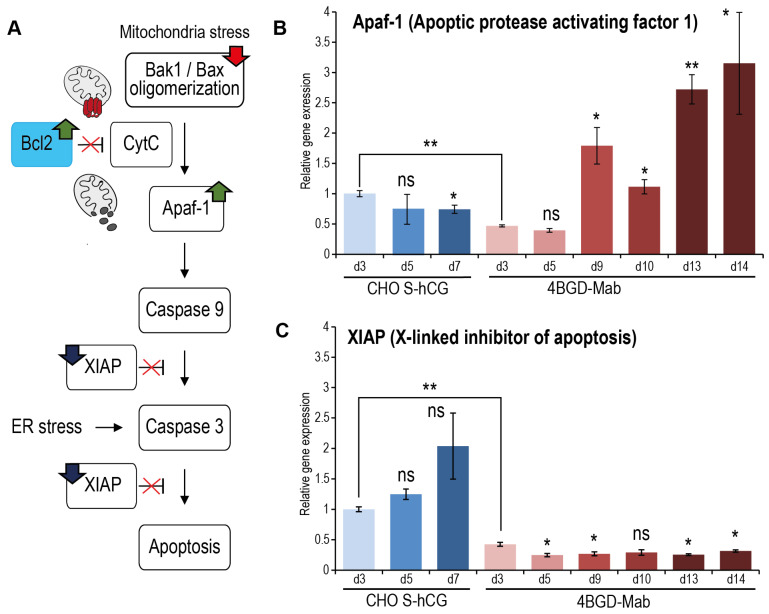
Diagram of the apoptosis induction pathway the RT-PCR analysis of the apoptosis-associated genes expression level. (**A**). Designations: Bak1—Bcl2 antagonist/killer 1; Bax—apoptosis regulator BAX; Bcl-2—apoptosis regulator B-cell lymphoma 2; CytC—Cytochrome c; Apaf-1—Apoptotic protease activating factor 1; XIAP—X-linked inhibitor of apoptosis protein. Green up arrows—increased expression level in 4BGD-mAb cells; blue down arrows—decreased expression level in 4BGD-mAb cells; red down arrow—gene inactivation in 4BGD-mAb cells. Quantitative PCQ results for Apaf1 (**B**) and XIAP (**C**), cell samples for RNA extraction were taken at days 3–14, designated as dN. Cell cultures are from Figure 2C. Error bars represent standard error of measurement, *n* = 3–4. Statistical significance of differences by the two-tailed *t*-test, *—*p* < 0.05, **—*p* < 0.01, ns—*p* > 0.05. Data points are compared with day 3 points, which are the same cell line if not marked by brackets.

**Figure 4 cells-14-00692-f004:**
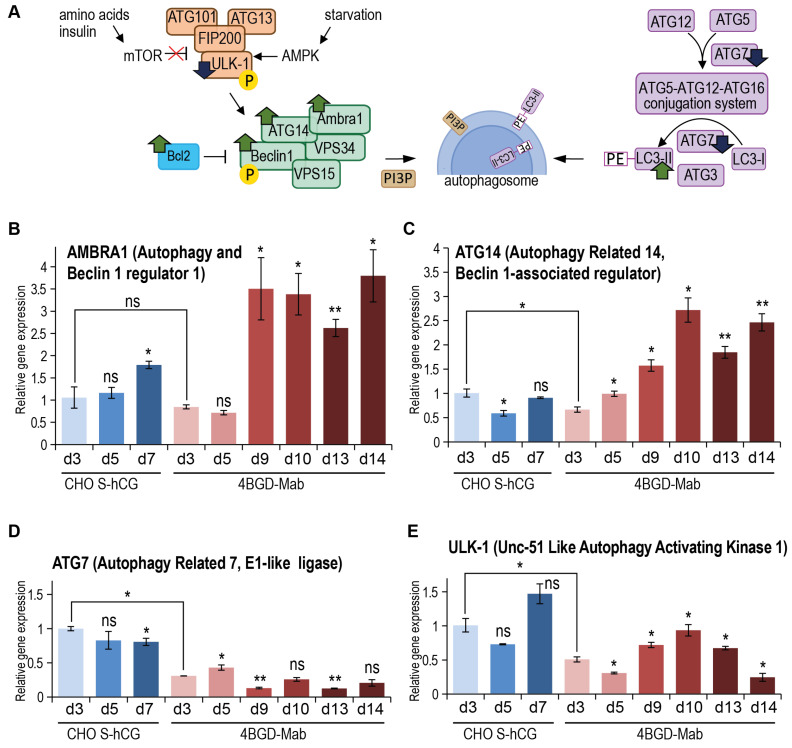
Diagram of the macroautophagy-related protein complexes and the RT-PCR of the macroautophagy-associated genes expression levels. (**A**): mTOR—mammalian target of rapamycin; ULK-1—UNK51-like kinase 1; ATG—autophagy-related gene; FIP200—focal adhesion kinase family interacting protein of 200 kD; AMPK—AMP-activated protein kinase; VPS15—vacuolar protein sorting 15, a serine/threonine protein kinase; VPS34—vacuolar protein sorting 34 phosphatidylinositol 3-kinase; AMBRA1—autophagy and beclin-1 regulator 1; PIP3—Phosphatidylinositol (3,4,5)-trisphosphate; Bcl-2—apoptosis regulator B-cell lymphoma 2; PE—Phosphatidylethanolamine; LC3- Microtubule-associated protein light chain 3. Green up arrows—increased expression level in 4BGD cells; blue down arrows—decreased expression level in 4BGD cells. Quantitative PCR data—panels (**B**–**E**). Cell samples for RNA extraction were taken at days 3–14, designated as dN. Cell cultures are from the Figure 2C. Error bars represent standard error of measurement, *n* = 3–4. The statistical significance test is the same as Figure 3. **p* < 0.05, ***p* < 0.01, ns: no significance.

**Table 1 cells-14-00692-t001:** Primers and probes used for PCR analysis.

Primer	Nucleotide Sequence 5′ → 3′
RT-Mmadhc-F	TGTCACCTCAATGGGACTGC
RT-Mmadhc-R	CAGGTGCATCACTACTCTGAAAC
RT-bACT-F	GCTCTTTTCCAGCCTTCCTT
RT-bACT-R	GAGCCAGAGCAGTGATCTCC
RT-XIAP-R	TTTACACCCTGGGTACCACT
RT-XIAP-F	GAGGAGGGCTCACAGATTGG
RT-APAF1-R	TGTCAAACATGAGCCAAGCC
RT-APAF1-F	CTGCCTAAGGCGAGCGATTA
RT-Ambra1-F	CAGTCTGCTGTGGCCAGTAA
RT-Ambra1-R	TCACGGAGGCATTGCTGATT
RT-ATG14-58F	GCTTCTTCTGTTGGTAGACTTG
RT-ATG14-58R	CGAGGTGGAGGATATAGAGTTAG
RT-ATG7-58F	AAGGCTGGCTGAGTCATC
RT-ATG7-58R	TGTCCAAGTCCAAGGTAGG
RT-ULK1-58F	CAAGAAGAACCTCGCCAAGTC
RT-ULK1-58R	GGAAGTCATACAACGCCACAA
TA-DHFR-1v7-F	GGGGACCCTGGTCACGTGTGCT
TA-DHFR-1v7-R	GCCTGGTGGTGGAGGCGGAGTCT
TA-GS-2v4-F	TCTTCGTGTTTGTCATAAGCC
TA-GS-2v4-R	TAACTGGGCACGAGGAATAAA
SQ-BAX456-F	TGTCTCCCTCGTAGCCCCTAT
SQ-BAX456-R	AGCCTTGCTTGTTTTGTTCG
SQ-BAK467-F	CTGTGCTCTTGGTTTCTTTCACG
SQ-BAK467-R	AGGGGTGGGTTACAGAGTGGC

**Table 2 cells-14-00692-t002:** Potential off-target editing events in the CHO 4BGD cells.

Gene Name	gRNA Sequence	Number of Potential Off-Target Sites	Off-Target Mutations Found	Potentially Affected Genes
Bak1	GGAAGCCGGTCAAACCACGT	158	2	ABCC4
Bax	GCTGATGGCAACTTCAACTG	39	3	CUNH12orf56, Nepro
Glul	GGCCTCCTCGATGTGCCTGG	86	2	Cep162 (heterozygous)
DHFR	GCAAGAACGGAGACCTTCCC	65	4	Sumf1
DHFR	GGTGATCGCTGCTGCTGTCA	120	3	-

**Table 3 cells-14-00692-t003:** Copy numbers of the integrated plasmids in the CHO 4BGD cells.

DNA Region	ORF	Mean Coverage	Calculated Copy Number per Diploid Genome
CHO genome	-	9.97527	1
p1.2-Zeo-Bcl2 plasmid	Bcl-2	5.12919	1.0
p1.2-Zeo-Bcl2 plasmid	Zeocin resistance	8.85867	1.8
p1.2-Hygro-Beclin1 plasmid	Beclin1	10.3415	2.1
p1.2-Hygro-Beclin1 plasmid	Hygromycin resistance	20.769	4.2
*Bcl2* (Gene ID: 100774873)	-	9.76513	1.0
*Becn1* (Gene ID: 100762045)	-	9.46119	1.0

## Data Availability

Complete new-generation sequencing data have been deposited in the Sequence Read Archive (SRA) under the accession number SRR24907270 (BioProject: PRJNA983294). Cell line CHO 4BGD is deposited to the National Bioresource Center Russian Collection of Industrial Microorganisms, Kurchatov Institute Research Center (https://vkpm.genetika.ru/), cat. H-233, and is freely available.

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
