# Peer review of "Genomic and Phenotypic Characterization of CHO 4BGD Cells with Quad Knockout and Overexpression of Two Housekeeping Genes That Allow for Metabolic Selection and Extended Fed-Batch Culturing"

_cells, 2025, doi:10.3390/cells14100692_

Round 1
Reviewer 1 Report
Comments and Suggestions for Authors
Using Crispr/Cas9, the authors generated CHO S cells deficient in pro-apoptotic and selection markers and overexpressing Beclin1 and Bcl-2. 10x read depth genome sequencing was peformed and possible off-target effects were examined; the data indicate that most likely no other genes were functionally affected by off-target events. Also the copy number of the transfected Bcl2 and Beclin-1 cDNAs were determined from the genome sequencing data. Thus, the study provides an in-depth analysis of the genomic alterations in the generated cell line. The generated line as then used for expression of the humanized omalizumab antibody and its properties compared to a parental CHO-S cell expressing hCG.
Comments
In order to demonstarte a significant improvement by the method developed in this study (compared to not engineered cells), it would have been better to compare the new modified cell line with a parental CHO-S cell expressing the same protein/antibody.
Along this line; can the authors exclude that the hCG expressing clone (which if I understand it correctly was not sequenced) did not show genomic alterations that potentially could explain reduced viability after longer growth time? A direct comparison of the genomes of the two lines would have been more appropriate, instead of comparing the 4BGD genome to the reference genome in the data base.
It would also be more informative, if the engineered line would be examined with and without Beclin1/Bcl2 overexpression, to examine whether really all genetic changes are necessary and/or which alterations have the strongest effect.
Are the time-dependent changes in gene expression and the difference between CHO-S-hCG and 4BGD at day3 and day5 statistically significant? An appropriate statistic test should be performed to demonstrate that this the case (or not).
Author Response
We thank Reviewer 1 for their thorough and constructive review, which has significantly improved our manuscript.
Comment 1: In order to demonstrate a significant improvement by the method developed in this study (compared to not engineered cells), it would have been better to compare the new modified cell line with a parental CHO-S cell expressing the same protein/antibody.
Response: We fully agree that parallel producer strains would provide an ideal comparison. However, due to resource limitations, we utilized two established industrial-grade clonal cell lines (each selected from hundreds of candidates for commercial partners). While we employed only short-term culturing for clone selection, the fed-batch longevity assessment was equitable. As previously demonstrated, the 4BGD parental line shows substantially enhanced longevity in batch culture. We have now included growth curves for both CHO-S and 4BGD cell lines in Eden medium (fed-batch culture), which confirm consistent growth patterns.
Comment 2: Along this line; can the authors exclude that the hCG expressing clone (which if I understand it correctly was not sequenced) did not show genomic alterations that potentially could explain reduced viability after longer growth time?
Response: We performed additional fed-batch cultures of parental CHO-S and 4BGD lines in Eden S600 medium (new Fig 2D). Both parental lines exhibited greater longevity than their protein-secreting descendants, with CHO-S cultures terminating by day 15 while 4BGD remained viable (>19 days; still viable at resubmission). The CHO S-hCG line demonstrated longevity comparable to parental CHO S, showing no signs of premature culture collapse. NGS analysis of this line (manuscript in preparation) revealed no significant genomic deviations from the CHO reference genome. The transgenes were integrated as plasmid concatemers in chromosome 2, outside transcribed regions.
Comment 3: A direct comparison of the genomes of the two lines would have been more appropriate, instead of comparing the 4BGD genome to the reference genome in the data base.
Response: We agree that comprehensive sequencing of both engineered and parental cells would provide optimal characterization. In this study, we demonstrate that multiplex CRISPR/Cas9 editing with plasmid-encoded gRNAs produced no detectable off-target events (with sufficient sequencing depth for all target loci analysis). Future work will include Nanopore sequencing for complete 4BGD genome characterization.
Comment 4: It would also be more informative, if the engineered line would be examined with and without Beclin1/Bcl2 overexpression, to examine whether really all genetic changes are necessary and/or which alterations have the strongest effect.
Response: We previously addressed this in our Russian publication https://sciencejournals.ru/cgi/getPDF.pl?jid=biotekh&year=2022&vol=38&iss=4&file=BioTekh2204008Kovnir.pdf . Figure 2 (red line, "10/22" cell line) shows growth curves without Beclin1/Bcl2 overexpression (negative control: "A11" CHO-S derivative). Our data indicate Beclin1/Bcl2 overexpression affects peak cell density but not longevity. We followed the original work of Jae Seong Lee (DOI: 10.1002/bit.24879) there both proteins had to be overexpressed for maintaining the normal peak cell density and increased culture duration. It is possible that Bak1/Bax knockout cells do not need the Bcl-2 overexpression for active autophagy-mediated survival in the aged culture, we plan to find the needed level of Beclin-1 and Bcl-2 expression levels in further studies by generation of many clonal cell lines, overexpressing only Beclin-1 or both proteins. We added the following sentence to the Introduction to clarify the behavior of 10/22 and 4BGD cells, as it was reported by us in Biotechnology(Rus) journal: “Unlike the parental 10/22 cell line, it achieved the same peak cell density as intact CHO-S cells while maintaining longevity comparable to the 10/22 line (unlike CHO-S).”
Comment 5: Are the time-dependent changes in gene expression and the difference between CHO-S-hCG and 4BGD at day3 and day5 statistically significant? An appropriate statistic test should be performed to demonstrate that this the case (or not).
Response: We have repeated the Beclin-1 blotting and found that I (the corresponding author) made a mistake in membrane image processing. I have erroneously flipped it horizontally because I was thinking that Beclin-1 is changed over time approximately in the same way to the Bcl-2. In fact, Beclin-1 is going down over time. We made two more biological repeats for Beclin-1 and Bcl-2 blots, raw membrane images are added to the Supporting Materials. The Beclin-1 is downregulated over time, the Bcl-2 is going up. I suppose that there is no need for the statistical analysis of the Beclin-1 dynamics, since both cell lines apparently downregulate it at the plateau phase. We updated the Fig. 2 with new blotting data and made several changes in the Results and Discussion sections, withdrawing any claims of the Beclin-1 upregulation in the aged culture of 4BGD cells
Reviewer 2 Report
Comments and Suggestions for Authors
In their article Orlova et al. described and functionally characterized genetically modified producer CHO 4BGD cell line. Authors provided detailed characterization of introduced changes in genomic sequence and potential consequences of CRISPR/Cas9-mediated editing. Finally, they also provided a functional characterization of the producer cell line and confirmed its superiority to currently available alternatives.
Experiments are designed properly, methods are fully described, results are presented in a clear way and are properly discussed.
I have following comments:
1) As the exact characterization of genetic changes is one of the main focuses of this paper, I would suggest to characterize in more details the transfected (integrated) transgenes. For example, the determination of copy number (section 3.3.) is not much clear to me. In my opinion, it should be possible to precisely characterize the coverage of transgenes compared to endogenous variants as transgenes do not contain introns and thus potentially specific reads should be present for both “only” endogenous and “only” transgene sequences.
Moreover, as authors used pair-end sequencing, I would guess it should be also possible to determine the correct insertion site(s) for both transgenes. May be that by analyzing reads for which only one in the pair maps to transgene(s) would enable such determination.
It would be important to know the exact localization of insertions as they could potentially disrupt some endogenous genes.
Did authors perform such analysis? Could authors discuss this in more details?
2) Authors claim that Beclin-1 expression is increased in response to environmental stress (e.g. lines 653-654 in Discussion). However, considering the transgenes are under control of EEF1A1 promotor, their regulation will no more be connected to autophagy anymore. Based on data presented in Fig. 2D, it seems to me that the only difference is the absence of apoptosis-mediated Beclin-1 cleavage in 4BGD cell line (which in turns enable proper/stronger autophagy activation).
Bcl-2 expression seems to behave the same in both 4BGD and CHO S lines, indicating that the changes are mediated probably from the endogenous locus and transgene provides only increased background expression.
In case of Beclin-1 the regulation might be similar.
It is probably not possible now, but I would suggest to analyze by WB also EF1A1 protein by itself to get information about regulation of this promoter during grown/production phase.
And it would be also valuable to use (in "next" version of the cell line) tagged transgenic proteins to distinguish their expression (and regulation) from endogenously produced variants or apoptosis-resistant (Caspase 3 cleavage-resistant) Beclin-1 variant (if possible to construct).
Minor points:
1) Fig. 1D is referenced in the text before Figs. 1B and 1C. It is a little bit confusing and I would suggest to correct the Figure accordingly.
2) All figures are slightly blurred – I would suggest to provide higher-quality pictures for the final version.
3) Line 380: what does “mln” means?
4) Line 482: Fig. 2B is referenced here, but in the figure, there are no data about viability which is mentioned in line 481. Please, either clarify the text or correct the figure(s).
Author Response
We thank the Reviewer 2 for the informative and rigorous review and for valuable advices. The questions raised significantly improved the article.
Comment 1: As the exact characterization of genetic changes is one of the main focuses of this paper, I would suggest to characterize in more details the transfected (integrated) transgenes. For example, the determination of copy number (section 3.3.) is not much clear to me. In my opinion, it should be possible to precisely characterize the coverage of transgenes compared to endogenous variants as transgenes do not contain introns and thus potentially specific reads should be present for both “only” endogenous and “only” transgene sequences.
Response: We made additional mapping of sequencing data to the plasmids inserted and updated the Results section, subsection 3.3 and Table 2.
Comment 2: Moreover, as authors used pair-end sequencing, I would guess it should be also possible to determine the correct insertion site(s) for both transgenes. May be that by analyzing reads for which only one in the pair maps to transgene(s) would enable such determination.
Response: For transgene detection, we followed methodologies from (1) https://doi.org/10.1186/s12896-017-0386-x and (2) https://doi.org/10.1080/09168451.2018.1506312. However, the plasmids used by us contain non-unique sequences (large fragments of the EEF1A1 CHO gene), resulting in unmapped reads distributed evenly across all chromosomes, which corresponds to random alignments. We used following pipeline for determination of the number of paired-reads, which mapped to 2 different chromosomes:
“samtools view -h Aligment.bam | awk '($3!=$7 && $7!="=")' | samtools view -Sbh > diffchr.bam
samtools coverage diffchr.bam > 4BGD_diffchr.csv”
The output is as follows:
Chromosome code |
Chromosome length |
Number of reads |
Coverage |
NC_048595.1 |
461620117 |
672437 |
0.138569 |
NC_048596.1 |
282827514 |
508367 |
0.171485 |
NC_048597.1 |
231097868 |
437548 |
0.180418 |
NC_048598.1 |
193770019 |
341411 |
0.167179 |
NC_048599.1 |
155611870 |
294807 |
0.180697 |
NC_048600.1 |
134359064 |
217798 |
0.153343 |
NC_048601.1 |
99554469 |
233368 |
0.223601 |
NC_048602.1 |
28505831 |
200118 |
0.658 |
NC_048603.1 |
32558357 |
126952 |
0.376296 |
NC_048604.1 |
127255434 |
206848 |
0.152154 |
The number of such reads is very similar for all chromosomes, we can’t deduce any real transgene insertion events (paired reads linking two chromosomes together) from these data.
The same analysis was performed for supplementary (chimeric) reads:
“samtools view -f 2048 -Sbh Aligment.bam > supplementary.bam
samtools coverage supplementary.bam > 4BGD_supplementary.csv”
Chromosome code |
Chromosome length |
Number of reads |
Coverage |
NC_048595.1 |
461620117 |
40070 |
0.22018 |
NC_048596.1 |
282827514 |
27690 |
0.242495 |
NC_048597.1 |
231097868 |
28134 |
0.273116 |
NC_048598.1 |
193770019 |
23922 |
0.307838 |
NC_048599.1 |
155611870 |
17892 |
0.268798 |
NC_048600.1 |
134359064 |
12554 |
0.225782 |
NC_048601.1 |
99554469 |
13639 |
0.319252 |
NC_048602.1 |
28505831 |
20312 |
1.11157 |
NC_048603.1 |
32558357 |
5149 |
0.414917 |
NC_048604.1 |
127255434 |
11928 |
0.210393 |
Resulting table shows similar distribution. The dataset obtained by us is insufficient for positioning of transgene on CHO chromosomes.
Comment 3: It would be important to know the exact localization of insertions as they could potentially disrupt some endogenous genes. Did authors perform such analysis? Could authors discuss this in more details?
Response: We agree that exact positioning of the target genes (Beclin-1 and Bcl-2) is important for the cell line characterization. The insertion sites for Beclin and Bcl-2 were not detected, as no relevant paired reads were found. Unlike in human studies, CHO cells lack an annotated mutation database, making it impossible to definitively predict whether a given substitution would lead to gene function loss.
Comment 4: Authors claim that Beclin-1 expression is increased in response to environmental stress (e.g. lines 653-654 in Discussion). However, considering the transgenes are under control of EEF1A1 promotor, their regulation will no more be connected to autophagy anymore. Based on data presented in Fig. 2D, it seems to me that the only difference is the absence of apoptosis-mediated Beclin-1 cleavage in 4BGD cell line (which in turns enable proper/stronger autophagy activation).
Response: We would like to specifically thank the Reviewer for this comment. We made additional Beclin-1 blotting experiments, aiming to add data for statistical analysis and have found that initial blot image was erroneously flipped horizontally, so the time-dependent increase of Beclin-1 was a mistake. It is, in fact, decreased over time for both CHO S and 4BGD. We made 2 mutually independent biological repeats, both of them show the same pattern – Beclin-1 decrease over time. We updated the Figure 2 and article text accordingly.
Comment 5: Bcl-2 expression seems to behave the same in both 4BGD and CHO S lines, indicating that the changes are mediated probably from the endogenous locus and transgene provides only increased background expression. In case of Beclin-1 the regulation might be similar.
Response: We agree that Bcl-2 levels changes drastically over time for both cell lines and this excursion is not connected to the added Bcl-2 gene. We introduced Bcl-2 to the cells together with the Beclin-1 according to the previously published results. Perhaps the constantly overexpressed Bcl-2 is not needed at all in the Bak1/Bax knockout cells. We plan to check it in detail in future studies.
Comment 6: It is probably not possible now, but I would suggest to analyze by WB also EF1A1 protein by itself to get information about regulation of this promoter during grown/production phase.
Response: We would like to thank the Reviewer for this advice. We found a published article https://doi.org/10.1038/s41598-022-11342-1 where activity of the EF1A promoter in CHO cells was already measured over fed batch culture by the RNAsec. Thirty percent drop was seen during 14 days run. This change can’t explain the Bcl-2 and Beclin-1 change over time in our cultures.
Comment 7: And it would be also valuable to use (in "next" version of the cell line) tagged transgenic proteins to distinguish their expression (and regulation) from endogenously produced variants or apoptosis-resistant (Caspase 3 cleavage-resistant) Beclin-1 variant (if possible to construct).
Response: Yes, we totally agree that Bak1/Bax knockout cells should be re-engineered for better autophagy-mediated survival during fed batch runs. Most probably, c-myc or FLAG-tagged proteins should be used, utilizing one plasmid with IRES and/or 2A peptides. This research is already under way.
Comment 8: And it would be also valuable to use (in "next" version of the cell line) tagged transgenic proteins to distinguish their expression (and regulation) from endogenously produced variants or apoptosis-resistant (Caspase 3 cleavage-resistant) Beclin-1 variant (if possible to construct).
Response: Yes, we totally agree that Bak1/Bax knockout cells should be re-engineered for better autophagy-mediated survival during fed batch runs. Most probably, c-myc or FLAG-tagged proteins should be used, utilizing one plasmid with IRES and/or 2A peptides. This research is already under way.
Minor comment 1: Fig. 1D is referenced in the text before Figs. 1B and 1C. It is a little bit confusing and I would suggest to correct the Figure accordingly.
Response: We changed the panels order on Fig. 1 and added 2 panels there. Now the panels are referenced in the text correctly, from A to F.
Minor comment 2: All figures are slightly blurred – I would suggest to provide higher-quality pictures for the final version.
Response: It is a constant problem for us in MDPI journals – we upload hi-res pictures and Reviewers receive the blurred downsampled image data. Unfortunately, we can’t change it. We made the copies of all аigures in PDF and PNG formats in the shared folder - https://disk.yandex.ru/d/hel9nxIsbkJpPA .
Minor comment 3: Line 380: what does “mln” means?
Response: “mln” means “millions”. We corrected it to the “3*105 cells/ml”
Minor comment 4: Line 482: Fig. 2B is referenced here, but in the figure, there are no data about viability which is mentioned in line 481. Please, either clarify the text or correct the figure(s).
Response: Corrected to “Fig 2 A,B“. Viability is on panel A, titer – on panel B.
Round 2
Reviewer 1 Report
Comments and Suggestions for Authors
Critical points have to some extent addressed by the authors. That some points have not been addressed seems to be because of "resource limitations" as the author stated in their reply. This may be accepted as justification.
Nevertheless I strongly recommend to add a statstical analysis of the data in Figure 3 to demonstrate statistically significant effects (to comapre the two loines but also to see at which time point a significant increase in Apaf1 is found). Unfortunately in panel B, d5 for CHO S-hCG is missing (can the authors justify this?); it is therefore difficult to compare CHO S-hCG and 4BGD-Mab.
Author Response
We sincerely thank the Reviewer 1 for their insightful feedback. In response to the comments:
Comment 1: Nevertheless I strongly recommend to add a statistical analysis of the data in Figure 3 to demonstrate statistically significant effects (to compare the two lines but also to see at which time point a significant increase in Apaf1 is found). Unfortunately in panel B, d5 for CHO S-hCG is missing (can the authors justify this?); it is therefore difficult to compare CHO S-hCG and 4BGD-Mab.
Reply: We returned the missing data point to the Figure 3 B, it was excluded due to abnormal standard deviation in threshold cycles for both housekeeping genes chosen. We placed day 5 point back; it's similar to day 3 sample in Apaf-1 level and has large standard error of measurement. We added the t-test (unpaired, two-sided) analysis results to all diagrams on Figures 3 and 4 and updated figure's legends accordingly.
Also, we made a minor correction - added the day 20 data point to Figure 2 D, cell line 4BGD-mAb. Viability has dropped below 80% on day 20, so this culture was held to maturity. Supporting data file was updated to better represent all blots made.
Round 3
Reviewer 1 Report
Comments and Suggestions for Authors
The comments of my last review have been addressed appropriately.